# A Systematic Review of Job Demands and Resources Associated with Compassion Fatigue in Mental Health Professionals

**DOI:** 10.3390/ijerph17196987

**Published:** 2020-09-24

**Authors:** Jasmeet Singh, Maria Karanika-Murray, Thom Baguley, John Hudson

**Affiliations:** Department of Psychology, School of Social Sciences, Nottingham Trent University, Nottingham NG1 4FQ, UK; maria.karanika-murray@ntu.ac.uk (M.K.-M.); thomas.baguley@ntu.ac.uk (T.B.); john.hudson@ntu.ac.uk (J.H.)

**Keywords:** compassion fatigue, job demands, job resources, work-related factors, mental health professionals

## Abstract

Psychosocial hazards in mental healthcare contribute to the development of compassion fatigue in mental health professionals. Compassion fatigue has a negative impact on the mental health and wellbeing of professionals that can impair the quality of services provided to clients. The majority of research on compassion fatigue has focused on individual-level variables such as gender, history of trauma and age, among others. It is also imperative to understand the role played by alterable work-related characteristics in contributing to the development of compassion fatigue in order to attenuate its impact on mental health professionals and their clients. The present review examined articles exploring work-related factors associated with compassion fatigue. Fifteen quantitative studies were included and their quality was assessed using a checklist. An inductive content-analysis approach was adopted to synthesise the themes emerging from the data. The results suggested a theoretical model consistent with the Job Demands-Resources model, wherein job demands (such as workplace trauma, workload and therapeutic settings) are associated with compassion fatigue, and job resources (such as supervisors’, coworkers’ and organisational support) mitigate the impact of job demands. In addition to person-oriented factors, work-related factors are critical for the prevention of compassion fatigue.

## 1. Introduction

A vital skill for a mental health professional is to be able to empathise with clients [1]. Empathy is associated with the development of an effective therapeutic alliance with clients [2,3,4] which is associated with treatment outcomes [5,6]. A meta-analytic study by Elliott, Bohart, Watson and Greenberg [7] found that empathy is a significant predictor of the outcomes of psychotherapy. Rogers [2], a prominent advocate of the use of empathy in mental healthcare, defined it as the ability to enter the private world of the client as if it were one’s own without ever losing the “as if” quality. The proviso specified in this definition, i.e., of not losing oneself in the client’s narrative, may be easier said than done. To help alleviate a client’s pain or trauma, a mental health professional has to listen to, and to some degree absorb, the *pain of the client* [8,9]. However, recurrent exposure to distressing information may take a toll on the mental health of the mental health worker [10].

Excessive empathetic engagement with traumatic information is a risk factor for secondary traumatic stress [11,12]. It is a syndrome that mimics the symptoms of post-traumatic stress disorder (PTSD), but unlike PTSD, it is caused by secondary exposure to trauma [13,14]. Secondary traumatic stress has an aversive impact on the mental health and wellbeing of an individual, leading to psychological issues such as strained interpersonal relationships, insomnia (or poor sleep hygiene) and major depressive disorder [11]. If early measures to counter the negative effects of excessive empathic involvement are not taken, then the mental health professional might become vulnerable to burnout. Burnout involves a feeling of being emotionally drained, a sense of detachment towards the recipients of one’s care and a plummeting sense of competence [15]. Studies have highlighted moderate to high rates of burnout and secondary traumatic stress in mental health professionals such as clinical psychologists, counsellors, psychotherapists, psychiatric social workers and psychiatric nurses [16,17,18,19,20,21].

The combination of secondary traumatic stress and burnout is commonly referred to as *compassion fatigue* [13,22,23]. Although the term is used synonymously with *burnout* [15], *secondary traumatic stress* [23], *vicarious traumatisation* [24] and *traumatic countertransference* [9], the present review focuses exclusively on *compassion fatigue* for the purpose of consistency and because some researchers have attempted to distinguish it from other similar constructs, e.g., [10,25]. According to Cocker and Joss [26], compassion fatigue (CF) is the “*stress resulting from exposure to a traumatised individual. CF has been described as the convergence of secondary traumatic stress* (*STS*) *and cumulative burnout* (*BO*), *a state of physical and mental exhaustion caused by a depleted ability to cope with one’s everyday environment*.” (p. 1). The symptoms of compassion fatigue include chronic physical and emotional exhaustion, depersonalisation, feelings of inequity, touchiness, headaches, loss of weight and negative feelings toward work, life and others outside the therapeutic relationship [27]. Mental health professionals experiencing compassion fatigue might also experience self-contempt, feelings of low job satisfaction, psychosomatic ailments, absenteeism or substance abuse, among other adverse outcomes [28,29].

A 2013 cross-sectional study [30] reported that 70% of psychotherapists employed by UK’s National Health Service (NHS) were vulnerable to experiencing chronic levels of secondary traumatic stress and average levels of burnout. In addition, a 2016 report published by the British Psychological Society (BPS), based on a survey conducted by New Savoy, reported that 48% of psychotherapists in the NHS reportedly experienced symptoms of depression, 25% experienced a long-term chronic condition and 70% reported experiencing occupational stress [31]. Depression, chronic health impairment and work-related stress have all been found to be associated with compassion fatigue [11,28,29,32]. In spite of this, the existing literature has paid more attention to the role of relatively stable individual-level characteristics (such as gender, age, history of trauma, negative life events, coping style and attachment style, among others) [22,33,34,35,36,37,38,39,40,41,42,43], as opposed to more dynamic, extrinsic and potentially alterable work-related factors. A relatively small body of research that has investigated the association of work-related factors has found that workload, client-related difficulties, relationships with other health professionals, emotional labour and support from supervisors and/or co-workers are some of the work-related factors that have been found to be associated with compassion fatigue in mental health professionals [44,45,46,47,48,49].

A narrative review of factors associated with compassion fatigue in mental health professionals found that caseload (or the number of traumatised clients seen per week) was the only significant work-related factor predicting compassion fatigue in mental healthcare personnel [50]. Other nonwork-related factors included professionals’ history of trauma, trait-based mindfulness, empathy, socio-demographic factors (such as age, experience and gender) and religiosity [50]. However, it is important to highlight that the review focused on articles published until the end of August 2014. This implies the need for a revised, updated synthesis of studies, especially since the growing prevalence of various mental health conditions [51] could have an impact on the workload of the workforce in mental healthcare [52,53,54,55]. Furthermore, the review by Turgoose and Maddox [50] did not provide adequate information about reliability checks (such as the degree of inter-rater agreement for selecting articles for full-text review or for scores provided for assessing the quality of the studies), casting aspersions over the rigorousness of the study selection process and the transparency and replicability of the review [56,57]. According to the AMSTAR guidelines, a “good” review article should achieve an inter-rater agreement of at least 80% [58]. In contrast, the review by Turgoose and Maddox [50] did not provide any such information.

Considering compassion fatigue as an occupational health and safety issue, the literature in occupational health psychology suggests that providing primary interventions to eliminate stressors are more time- and cost-effective than delivering secondary interventions to develop personal resources [59,60]. For instance, amending the workload of professionals is generally preferred over influencing their psychological coping mechanisms. However, some evidence indicates that a combination of individual, group and organisational-level interventions are the most effective for the psychological issues faced by employees [61]. A systematic review of the literature on secondary interventions to reduce compassion fatigue in emergency and community service workers found that out of 13 included studies, only 4 (30%) reported a significant decrease in burnout and only 3 (23%) reported a significant decrease in secondary traumatic stress [26].

In light of the evidence presented above, it is important that compassion fatigue research places more emphasis on work-related factors for two primary reasons: (i) to complement the existing evidence on person-related factors; and (ii) to inform the development of ways to attenuate the impact of work-related factors on the mental health and wellbeing of mental health professionals. Therefore, the aim of the present systematic review is to synthesise and critically examine the existing evidence exploring the role of work-related factors in the development and mitigation of compassion fatigue in mental health professionals.

## 2. Materials and Methods

A review of the literature was conducted prior to the commencement of the systematic review. The results aided in the development of the research protocol which was agreed upon by the entire research team (JS, MKM, TB and JH). The systematic review process followed the guidelines of the Preferred Reporting Items for Systematic Reviews and Meta-Analyses (PRISMA) [62]. The PRISMA guidelines provide a set of items informed by empirical research for conducting systematic reviews and meta-analyses [62]. Although they were originally developed for reporting systematic reviews and meta-analyses collating evidence from randomised-controlled trails [62], they are now also being increasingly used for reporting systematic reviews of studies adopting varied research designs and in varied scientific disciplines, e.g., [19,26].

### 2.1. Search Strategy

Online databases to search relevant research studies included Web of Science, SCOPUS, PsycINFO, Science Direct and PubMed. The search strategy included a combination of the following terms: “*Compassion Fatigue*” AND (“*Clinical Psychologist*” OR “*Counsellor*” OR “*Counselor*” OR “*Psychotherapist*” OR “*Therapist*” OR “*Psychiatric Social Worker*” OR “*Mental Health Social Worker*” OR “*Psychiatric Nurse*” OR “*Community Psychiatric Nurse*” OR “*Community Mental Health Nurse*” OR *“Mental Health Professional”* OR *“Mental Health Worker”*). The search was restricted to studies published in English (or where an English translation was available) in peer-reviewed journals prior to 20 January, 2020. The reference lists of research studies were also manually examined for additional potential studies for inclusion.

The online and manual search of literature led to the identification of 806 unique articles. Following screening based on titles and abstracts, 150 articles were retained. Further, full-text review of articles to determine eligibility for inclusion in the present review identified 15 articles for qualitative synthesis (See Figure 1 for PRISMA flow diagram).

### 2.2. Inclusion and Exclusion Criteria

Studies that met the following criteria were included: (i) at least 50% of the sample population comprised of AMHPs (clinical psychologists, counsellors, psychotherapists, psychiatric social workers, psychiatric nurses or professionals employed in other-related allied mental healthcare services) working in organisational settings; (ii) the study examined work-related determinants and/or preventative factors of compassion fatigue using a quantitative methodology; and (iii) the study had analysed primary data (i.e., systematic reviews, meta-analytic reviews and narrative reviews were not included). In addition, no restriction was imposed on the research design adopted by a particular study. In cases of mixed-methods studies, only the quantitative results were included in the present review. Studies were eliminated if they examined medical practitioners (such as psychiatrists, nurses, physicians etc.) as their main cohort, despite those professionals being involved in the delivery of mental health services, owing to the differences in their training and philosophical approach to mental healthcare [63]. Also, quantitative studies that did not use validated measures of compassion fatigue were not included in the current review.

### 2.3. Study Selection, Data Extraction and Risk of Bias (Quality) Assessment

In the present review, following the PRISMA guidelines [62], postidentification of research articles and removal of duplicates, the first independent reviewer (JS) screened the titles and abstracts to determine the eligibility of the articles for inclusion in the present review. This was followed by an evaluation of full-text articles for assessment of eligibility. In case of a doubt regarding eligibility, the opinion of the third independent reviewer (TB) was sought. All disagreements were resolved through discussions.

The first step was succeeded by an extraction and tabulation of the characteristics of the included articles. A data extraction form was developed to standardise the processes of data extraction and qualitative synthesis. The form was peer-reviewed, with the feedback integrated into its further development. The final version of the data extraction form related to the following information from full-text articles: (i) country; (ii) sample size, sociodemographic characteristics and professional specialisation(s) of participants; (iii) aim(s) of the study; (iv) study design; (v) measure of compassion fatigue; (vi) dimensions of compassion fatigue measured (burnout and/or secondary traumatic stress); (vii) theoretical framework (if included); (viii) results of the study; and (ix) limitations of the study.

Two methods were adopted for the risk of bias (quality) assessment of the included studies. Firstly, only authenticated databases (as mentioned in Section 2.1. *Search Strategy*) were used to identify relevant articles. This enabled the researchers to identify articles published in peer-reviewed journals only. Secondly, each included study (i.e., full-text article) was assessed for its quality using the Crowe Critical Appraisal Tool (CCAT Version 1.4) [64]. CCAT Version 1.4 provides a checklist for appraising a research article for its methodological quality and empirical rigour [64]. It has well established inter-rater reliability [65,66] and construct validity [67], and is widely used in systematic reviews [68,69,70]. Following the guidelines of CCAT Version 1.4 [64], each included article was examined by an independent reviewer to assess the quality of its preliminaries (title and abstract), introduction, design, sampling, data collection, ethical matters, results and discussion. Each article was assigned a score out of a possible maximum of 40. All the articles were firstly assessed by the first reviewer (JS) and then a small subset, i.e., 33.33%, of those articles (*n* = 5) were randomly selected and evaluated independently by the second, third and fourth reviewers (MKM, TB and JH) against the inclusion criteria. Interrater agreement was observably strong: *k* = 0.83 (95% CI 0.70–0.97), 0.80 (95% CI 0.66–0.94) and 0.83 (95% CI 0.69–0.97). However, since Cohen’s Kappa is an omnibus index of agreement that is also vulnerable to chance-related inflation or bias [71], the present review calculated the proportions of specific agreement [71,72] to obtain additional evidence for the degree of interrater agreement. The proportion of positive agreement [0.76 95% CI (0.56, 0.96), 0.7 95% CI (0.47, 0.93), 0.55 95% CI (0.19, 0.90)] indicated a moderate degree of agreement and the proportion of negative agreement [0.87 95% CI (0.76, 0.98), 0.85 95% CI (0.73, 0.97), 0.89 95% CI (0.80, 0.99)] demonstrated a strong degree of agreement between the ratings for title and abstract screening provided by the first and three independent reviewers. This was followed by calculating intraclass correlation (ICC) using the Statistical Package for Social Sciences Version 24 (SPSS V.24, IBM, Armonk, New York, USA) [73] based on a mean rating (*k* = 3), two-way, mixed effects, consistency multiple raters model [74]. Although the two-way mixed effects model has limited generalisability, it was still used for the present review because the reviewers were not randomly selected [74].

### 2.4. Strategy for Data Synthesis

An inductive content analysis [75] approach was employed in order to identify the themes emerging from the narrative data. The rationale behind adopting an inductive content analysis approach, as opposed to a deductive approach, was that the former allows the development of a model or theory based on interrelationships among the themes emerging from the data [75]. In contrast, a deductive approach aims to examine whether the emerging data are consistent with a priori assumptions or established theories or models [75]. In the present review, the coding process outlined by Thomas [75] was adapted for qualitatively synthesising the predictors and preventive factors emerging from the existing literature. Following multiple readings of the identified, significant variables, the researcher (JS) organised the variables related to each-other into themes and subthemes (See Figure 2 for a brief overview).

## 3. Results

### 3.1. Characteristics of Included Studies

The characteristics of the included studies (*n* = 15) are summarised in Table 1. All these studies adopted a cross-sectional, quantitative research design. They represented research conducted in 10 countries. The majority of the studies (*n* = 6) were conducted in the United States [76,77,78,79,80,81]. However, one study was multisite, and thus, it recruited participants from three different countries viz. Germany, Austria and Switzerland [82]. Samples comprised professionals from diverse occupational groups such as counsellors [83], mental health social workers [76,78,79,80], mental health social work doctoral students [81], mental health nurses [84,85,86,87,88] employee assistance professionals [77], correctional officers [85], psychotherapists [89] and frontline mental health professionals [90]. The 15 included studies represented a combined sample size of 3356 participants (*M* = 239.7, *SD* = 143), with individual study sample sizes ranging from 36 [85] to 532 [80]. Two studies [76,78] used the same dataset, and thus, their data were included only once. Of the nine studies that reported the mean age of participants [77,79,80,83,84,85,87,89,90], the combined average age was 47.1 years (*SD* = 7.5 years) and ranged from 36.9 years to 59.9 years. In terms of sex distribution, the total number of female participants across all included studies was 1586 (47.3%). In all but one study [84], the percentage of male participants exceeded the percentage of female participants.

Five studies (*n* = 886) [77,81,83,84,85]] reported the prevalence rates for high, medium and low levels of secondary traumatic stress/compassion fatigue and burnout in their respective samples. The combined averaged prevalence rates indicated that 6.6% (*n* = 58) reported high levels of burnout, 6.8% (*n* = 60) reported medium levels of burnout and 10.8% (*n* = 96) reported low levels of burnout. This suggests that a total of 24.2% (*n* = 215) of participants in five studies experienced the effects of burnout. For secondary traumatic stress, 1.8% (*n* = 16) reported high levels of secondary traumatic stress, 7.4% (*n* = 65) reported medium levels of secondary traumatic stress and 10.6% (*n* = 94) reported low levels of secondary traumatic stress. Thus, 19.8% (*n* = 175) of participants in five studies reportedly experienced the symptoms of secondary traumatic stress. A comparison of the prevalence rates clearly shows that burnout emerged as the most commonly reported dimension of compassion fatigue by mental health professionals.

### 3.2. Quality Assessment of Studies

Table 1 provides CCAT [64] quality assessment scores for each study included in the review. Although CCAT [64] does not provide a categorisation of scores, the present study classified the studies into “poor” (≤27), “average” (≥28), “good” (≥32) and “very good” (≥35) based on quartile ranks. Six studies [76,77,79,82,89,90] were classed as poor, five studies [78,83,85,87,88] were rated as average, two [81,86] were ranked as good and two [80,84] were graded as very good. In addition, inter-rater reliability was determined by calculating intraclass correlation (ICC) based on a mean rating (*k* = 3), two-way, mixed effects, consistency multiple raters model. The results indicated a moderate degree of agreement among the ratings provided by three researchers [74]. The average measure ICC was 0.57 with a 95% confidence interval from −1.15 to 0.953, *F* (4, 8) = 2.35, *p* = 0.14.

### 3.3. Characteristics of Compassion Fatigue Measures

All included studies used standardised self-report measures/scales/questionnaires to examine compassion fatigue in their respective occupational groups. Majority of the studies (*n* = 5) [81,83,85,87,88] used the Professional Quality of Life Scale Version 5 (ProQOL 5) [91]. ProQOL 5 includes three subscales: Secondary Traumatic Stress, Burnout and Compassion Satisfaction. Apart from the study by Butler, Carello and Maguin [81] that utilised only two dimensions of ProQOL 5 scale (burnout and compassion satisfaction), the remaining four studies administered all the three subscales. The internal consistency estimates of the subscales of ProQOL 5 averaged across five studies (four in case of secondary traumatic stress) were 0.82 for secondary traumatic stress (*n* = 4) and 0.71 for burnout (*n* = 5). These values are similar to the estimates provided by the test developer [84]: α = 0.81 (for secondary traumatic stress) and α = 0.75 (for burnout).

The remaining studies (*n* = 10) used previous versions of ProQOL such as Compassion Fatigue Scale-Revised [76,78], ProQOL [77,89], ProQOL-III [80,82,86], ProQOL-IV [79,84] and ProQOL-IV-R [90].

### 3.4. Theoretical Framework

Eight research articles [76,77,78,79,80,83,86,87,88,90] explicitly stated the theoretical framework adopted to conduct the study (See Table 1 for more details). Four among the eight articles [80,86,88,90] used Stamm’s Professional Quality of Life Framework (See Figure 3), which serves as the basis for the ProQOL 5 scale [91]. The remaining four studies adopted theoretical frameworks from personality psychology [83], social psychology [87], biopsychology [76] and counselling psychology [77]. However, none of the included studies examined compassion fatigue using a framework or model based in occupational health psychology literature. In addition, seven articles [78,79,81,82,84,85,89] made no reference to a theory or model guiding their empirical investigations.

### 3.5. Qualitative Synthesis of Themes

As mentioned in Section 2.3. *Strategy for Data Synthesis*, at the outset, the researchers aimed to conduct an inductive content analysis; however, the interrelations among the themes emerging from the data—workplace trauma, workload, therapeutic settings, support from supervisors and co-workers and organisational resources and support—were consistent with an existing model of occupational health and wellbeing, i.e., the Job Demands-Resources (JD-R) model [92]. Thus, it could be inferred that the present review, in practice, employed a hybrid form of inductive and deductive approaches to content analysis.

The JD-R model states that a combination of experienced high job demands and low job resources poses a potential threat to mental health and wellbeing, whereas a combination of high job demands and high job resources fosters personal and professional growth and psychological wellbeing [92]. These major tenets of the JD-R model [92] have been supported by several empirical studies [93,94,95] and recently by a meta-analytic review of longitudinal studies based on the model [96]. In the present review, the qualitative synthesis of data suggested that the work-related factors found to be associated with the development and mitigation of compassion fatigue in mental health professionals were in alignment with the definitions of job demands and job resources provided by the JD-R model [92].

#### 3.5.1. Job Demands Associated with Compassion Fatigue

According to the JD-R model [92], job demands refer to those “physical, social or organisational aspects of the job that require sustained physical or mental effort and are therefore associated with certain physiological and psychological costs” (p. 501). Workplace trauma [77,78,81,85,86,87,88], workload [79,80,84] and therapeutic settings [80,89] were the physical, social and organisational aspects of the job which, in the present review, were found to be the most commonly reported work-related factors to be associated with the development of compassion fatigue among mental health professionals. The following sections elaborate on the interactions among these job demands and compassion fatigue in mental health professionals.

##### Workplace Trauma

Experience of traumatic incidents in the workplace was the most frequently reported job demand found to be associated with compassion fatigue in mental health professionals [77,78,81,85,86,87,88]. Empirical investigations examining it have suggested that its impact is unevenly distributed between secondary traumatic stress and burnout. Six out of seven studies indicated that history of workplace trauma [85], perceived risk for the future [86], prediction of aggressive behaviour of clients [87] and stress emanating from engagement with traumatised clients [77,81,88] were predictive of burnout in mental health professionals. In contrast, secondary traumatic stress was predicted by only two factors: involvement in recovery efforts of large-scale traumatic events such as 09/11 terrorist attacks [78], and mental strain resulting from work with clients experiencing traumatic symptoms [77,81]. The latter was the only factor that significantly predicted both secondary traumatic stress and burnout [77]. However, it is imperative to note that its effect size for burnout (*F* = 24.38, *p* < 0.001) was stronger than that for secondary traumatic stress (*F* = 10.39, *p* < 0.01). In addition, the study’s sample comprised employee assistance professionals [77] who differ from other mental health professionals in terms of their training, approach, workplace settings and the nature of their workload.

##### Workload

Four studies [79,80,84,90] examined the association between workload and compassion fatigue in mental health professionals. An incongruence or a lack of fit between the expectations of a mental health professional and the job in terms of workload [90] was found to be negatively correlated with secondary traumatic stress, *r* = −0.45, *p* < 0.01. The association was particularly observed in cases of psychiatric nurses with quantitative aspects of workload (such as working hours and number of patients seen per nurse in the night shift) being predictive of burnout [84] and qualitative aspects (such as traumatised clients on caseload) being associated with secondary traumatic stress [79,80]. Unexpectedly, work on weekends was the only quantitative aspect that was not associated with burnout, being instead associated with secondary traumatic stress (*β* = −0.57, *p* < 0.01) [84].

##### Therapeutic Settings

The relationship between therapeutic settings (clinical settings, therapeutic orientation of professionals) and compassion fatigue in mental health workers was investigated by only two studies [80,89] included in the present review. The role of clinical settings was uncovered by a cross-sectional study by Craig and Sprang [80] that found that mental health professionals working in inpatient care settings reported higher levels of burnout and secondary traumatic stress than their counterparts in community mental health centres and private practice. Another study reported that psychotherapists with current therapeutic practice in cognitive-behavioural therapy were more likely to experience burnout (*r* = 0.20, *p* < 0.001) than professionals with other varied therapeutic orientations [89]. The association between current therapeutic practice and secondary traumatic stress was not found to be statistically significant [89].

#### 3.5.2. Job Resources Associated with Compassion Fatigue

According to the JD-R model [92], job resources refer to those “physical, psychological, social, or organisational aspects of the job that may […] be functional in achieving work goals, reduce job demands and its related costs, or stimulate personal growth and development” (p. 501). In the present review, the job resources found to abate the impact of compassion fatigue in mental health professionals included support from co-workers [83,84,85,87], support from supervisors [84,85,86] and organisational resources and support [76,77,78,80,82]. The following sections elaborate on the associations between job resources and compassion fatigue in mental health professionals.

##### Support from Co-Workers

Support from colleagues or co-workers in the form of congenial relationships [*F* (4, 169) = 9.64, *p* < 0.001] [84], collaborative effort [*F* (4, 169) = 3.76, *p* < 0.05] [84], emotional support (*IRR* = 0.93, *p* < 0.05) [85], perceived competence (of staff) to cope with patient-aggression (*r* = −0.22, *p* < 0.01) [87] and a sense of belongingness in the workplace (*β* = −0.35, *p* < 0.001) [83] were found to be negatively associated with burnout. Emotional support from colleagues was the only job resource that was found to assuage the effects of secondary traumatic stress (*IRR* = 0.87, *p* < 0.001), in addition to those of burnout, in a cross-sectional study on psychiatric nurses in Greece [85].

##### Support from Supervisors 

Support from the supervisor or the line manager emerged as a job resource that can alleviate the symptoms of burnout and secondary traumatic stress in two studies included in the present review [85,86]. Support from the line manager (*IRR* = 0.90, *p* < 0.01) [85] and regular supervision (*IRR* = 0.87, *p* < 0.001) [85] were positively associated with low levels of burnout. In addition, trust between the supervisor and the employee [*β* = −1.2 95% CI (−2.038, −0.382)] [86] was found to be negatively associated with burnout. Similarly, support from the line manager (*IRR* = 0.92, *p* < 0.05) [85] and consultation with the line manager (*IRR* = 0.92, *p* < 0.05) [85] were found to mollify the effects of high levels of secondary traumatic stress in mental health professionals.

##### Organisational Resources and Support

The provision of organisational resources and support emerged as the most frequently reported job resource assuaging the development of compassion fatigue in mental health professionals [76,77,78,80,82]. The studies included in the present review indicated that mental health professionals who availed the services provided by employee assistance professionals in their organisations (*F* = 4.01, *p* < 0.05) [77] and/or received customised training to work with traumatised clients, (*β* = −0.09, *p* < 0.05) [80] experienced lower levels of burnout than their colleagues. Also, mental health professionals employed at organisations that provided adequate information to work effectively with traumatised clients reported lower levels of burnout and secondary traumatic stress [76,78].

Since both quantitative [97] and qualitative evidence [98] suggest that healthcare professionals’ use of evidence-based practices is contingent upon the values that their organisation espouses, the present review included mental health professionals’ use of evidence-based practices under *Organisational Resources and Support*. A cross-sectional study [80] found that clinical psychologists’ and clinical social workers’ use of evidence-based practices inoculated them against burnout (*β* = −0.12, *p* < 0.01) and secondary traumatic stress (*β* = −0.09, *p* < 0.05). In addition, a multisite study on trauma therapists [82] reported that therapists who advocated working through trauma but did not practice it experienced higher levels of secondary trauma than those who advocated as well as practiced it, *F* (2, 91) = 4.84, *p* < 0.01. Further, the degree of working through trauma was negatively associated with burnout (*r* = −0.21, *p* < 0.05) [82].

### 3.6. Additional Findings

#### 3.6.1. Association between Secondary Traumatic Stress and Burnout

Five studies [81,82,83,84,88] reported a correlation between secondary traumatic stress and burnout. The relationship examined in four studies [82,83,84,88] using the Professional Quality of Life Scale (ProQOL), was found to be positive and ranged from medium (*r* = 0.47, *p* < 0.01) [88] to strong (*r* = 0.819, *p* < 0.001) [82] in degree. This reflects the shared variance between the burnout and secondary traumatic stress subscales of ProQOL 5 [91]. Only in one study [82] was the degree of correlation between the two constructs very strong. The remaining studies identified a correlation of moderate strength only, similar to the validation study by Stamm [91], *r* = 0.58. One study measured secondary traumatic stress using the Secondary Traumatic Stress Scale [99] and found its association with burnout subscale of ProQOL 5 [91] to be moderate in strength (*r* = 0.62, *p* < 0.01) [81]. In another cross-sectional study [90], scores on the secondary traumatic stress subscale of ProQOL 5 [91] were correlated with the three dimensions of burnout measured using the Maslach Burnout Inventory (MBI) [100]. The results suggested that the relationship between the two constructs was moderate in degree [90]: secondary traumatic stress and emotional exhaustion (*r* = 0.59, *p* < 0.01); secondary traumatic stress and cynicism (*r* = 0.39, *p* < 0.01); and secondary traumatic stress and reduced personal achievement (*r* = −0.21, *p* < 0.01).

#### 3.6.2. Compassion Satisfaction

Compassion satisfaction refers to the pleasure one derives from helping or assisting others [91]. Five studies included in the present review [77,81,83,84,85] reported the prevalence rates for high, average and low levels of compassion satisfaction. The combined sample from five studies represented 886 mental health professionals (*n* = 886). Among them, 5.33% (*n* = 47) reported high levels of compassion satisfaction, 10.1% (*n* = 89) reported average levels of compassion satisfaction and 4.8% (*n* = 42) reported low levels of compassion satisfaction. In total, 20.2% (*n* = 179) of participants experienced compassion satisfaction through their work in mental healthcare.

The relationship between compassion satisfaction and components of compassion fatigue (secondary traumatic stress and/or burnout) was also reported by some of studies included in the present review [81,83,84,88]. The negative relationship between compassion satisfaction and secondary traumatic stress was low in degree, with a range of −0.16 to −0.28 [81,83,84,90]. For compassion satisfaction and burnout, the relationship was found to be average in strength, with a range of −0.47 to −0.66 [81,83,84,88]. Also, the internal consistency estimate of compassion satisfaction subscale (α = 0.90) of ProQOL 5 [91] averaged across five studies [81,83,85,87,88] was similar to the original estimate (α = 0.88) reported by the test developer [91].

In addition, the job resources that were found to allay the effects of burnout or secondary traumatic stress were found to also promote compassion satisfaction in the same occupational group. The most frequently reported job resource was support from co-workers [82,83,84,85,86]. Emotional support from colleagues (*β* = 5.33, *p* < 0.001) [85] and a having a sense of community (*r* = 0.40, *p* < 0.01) [90] or belongingness (*β* = 0.35, *p* < 0.01) [83] at the workplace were found to promote compassion satisfaction among mental health workers. A cross-sectional study on Greek psychiatric nurses [84] indicated that nurses who reported that the staff always worked as a team (*M* = 32.9, *SD* = 9.4) experienced higher levels of compassion satisfaction than those who reported less frequent teamwork (*M* = 22.3, *SD* = 7.5), *F* (4, 169) = 7.40, *p* < 0.001. Support from the supervision or management was also found to reinforce a sense of satisfaction in mental health workers for helping their clients [85]. Lastly, the use of evidence-based practices was also found to escalate compassion satisfaction [82,85,89]. A study on US-based clinical psychologists and clinical social workers [80] showed that professionals with trauma training (*M* = 43.8, *SD* = 5.2) reported higher compassion satisfaction than professionals without it (*M* = 41.4, *SD* = 6.6), *t* (499) = −4.42, *p* < 0.001. Also, professionals with training and practice in transpersonal therapy were more likely to be satisfied with their work in mental healthcare [89].

## 4. Discussion

The aim of this systematic review was to systematically collate, synthesise and scrutinise existing empirical evidence investigating the role of work-related factors in the advancement and prevention of compassion fatigue in mental health professionals. A small but growing body of research has examined the association between work-related factors and compassion fatigue in mental health professionals [44,45,46,47,48,49]. The relationship between work characteristics and compassion fatigue has a deleterious impact on the physical and psychological wellbeing of professionals and the quality of services they deliver to their clients. In spite of this, studies exploring this construct have elucidated the role of relatively stable individual-level traits more than more readily alterable work-related characteristics. The review aimed to address this gap. It identified a moderate number of quantitative studies (*n* = 15) conducted across 10 different countries and published in peer-reviewed journals.

Five studies included in this review [77,81,83,84,85] reported the prevalence rates for secondary traumatic stress and burnout. The results indicated that burnout was the most commonly reported dimension of compassion fatigue in mental health professionals, with 24.2% of participants experiencing it, as opposed to 19.8% of participants reporting symptoms of secondary traumatic stress. In addition, the positive relationship between burnout and secondary traumatic stress reported by five studies [81,82,83,84,88] showed that the two constructs share common variance. This suggests that therapeutic work with clients who experience mental ailments depletes the emotional resources of professionals, leading to chronic stress which could further lead to the transfer of symptoms from the client to the professional. A longitudinal study on US military personnel and Polish human service workers confirmed that burnout precedes the development of secondary traumatic stress [25]. However, the study adopted a two-wave, cross-lagged panel design inhibiting the observation of curvilinear trends and confounding the observed changes with measurement errors [101,102,103]. Also, it is vital to note that the prevalence rates reported by the studies included in the current review used the 2010 cut-off criteria provided by Stamm [91], which have now been replaced by up-to-date normative data [104]. No study included in the present review used the recent cut-off scores to classify participants into low, average and high categories, thereby obfuscating the prevalence rate of compassion fatigue in mental health professionals. Despite such methodological shortcomings, the descriptive findings of the included studies highlight that compassion fatigue is a pertinent occupational health issue that necessitates a targeted approach to workplace intervention.

The information extracted from the studies was used to conduct a hybrid form of inductive and deductive content analysis (See Figure 2) to organise the themes emerging from the data into a framework. The themes were found to be in alignment with the Job Demands-Resources (JD-R) model [92], with job demands hampering the psychological health and wellbeing and job resources allaying the impact of job demands on the psychological health of mental health professionals. The job demands found to promote compassion fatigue in mental health workers included workplace trauma, workload and therapeutic settings. Mental health professionals working in in-patient care settings [80], with a high caseload of clients experiencing post-traumatic stress disorder (PTSD) [79,80,84,90], having training and current practice in cognitive-behavioural therapy [89] and being exposed to primary and/or secondary trauma at the workplace were more likely to experience compassion fatigue [77,78,81,85,86,87,88]. In particular, the number of clients, therapeutic orientation in cognitive-behavioural therapy and primary exposure to trauma at the workplace appear more likely to lead to burnout. In contrast, the nature of the caseload and secondary exposure to trauma appear likely to contribute to professionals’ secondary traumatic experiences. This suggests that *second-hand trauma*, an indirect exposure to trauma, has the potential to alter the life-script of a mental health professional. According to the constructivist self-development theory [24,105,106] which is often used to explain a similar phenomenon known as vicarious trauma, an individual develops cognitive schemas to interpret the experiences of his or her life [106]. However, the divulgence of a client’s traumatised narrative can alter a professional’s cognitive schemas or perceived realities [105]. A deep sense of empathy with a client’s traumatised narrative [107] can foster the development of an intimate therapeutic alliance wherein a professional might imitate a client’s overt and covert experiences to such an extent that the distinction between the professional and the client might cease to exist. Thus, in such a case, a professional might display the symptoms of a client and might experience secondary traumatic stress [107].

The experience of psychosocial hazards such as challenging clinical settings, direct exposure to trauma or the amount of quantitative workload are likely to exhaust a professional’s resources which could further promote affective, behavioural and cognitive withdrawal from the occupation or the clients. The role of increased workload or caseload [108,109,110,111] and challenging work settings (such as employment in community mental health centres) [112,113] in contributing to elevated levels of burnout has been replicated in previous research. According to the conservation of resources (COR) model [114], threat to personal resources (such as energy or individual traits) propel avoidance or escape behaviour to protect or conserve the existing, depleting repository of resources [114]. This suggests that compromised empathic resources could lead mental health professionals to distance themselves from their clients to safeguard their mental health and wellbeing, and to prevent a ripple effect of a dearth of resources on other areas of life. A cross-sectional study adopting the COR model [114] showed that differentiation of the self from the clients, specifically in social workers working with highly traumatised groups, was negatively associated with burnout, which was further negatively related to marital quality [115]. However, it should be noted that the study collected data using convenience sampling, thereby introducing potential sampling bias.

It is imperative to acknowledge here that the majority of the job demands examined by studies included in the present review were categorical in nature. The therapeutic orientation of professionals, clinical settings, experience of trauma at the workplace (yes/no), quantitative workload, number of traumatised clients on the caseload and involvement in recovery efforts for national tragedies inter alia are quantifiable, dichotomous variables. Although some of them, such as therapeutic practice or the number of clients on the caseload, can be quantified, the complexity of others, especially workplace trauma, could be measured more accurately using standardised self-report measures. Only one study included in the present review [88] measured exposure to physical and/or verbal violence at the workplace using a four-item scale developed by the research team. Although the scale demonstrated adequate internal consistency [88], its other psychometric properties such as factorial validity and norms remain unestablished in the literature. Studies outside the scope of this review [116,117,118] have examined mental health professionals’ traumatic experiences using standardised scales such as PTSD Checklist-Civilian Version (PCL-C) [119] or PTSD Checklist for *DSM–5* (PCL-5) [120]. No study included in the present review used either of the two checklists or some other standardised scale or questionnaire, thereby compromising the accurate measurement of workplace trauma.

According to the JD-R model [92], job resources mitigate the effect of job demands on employees’ physical and mental health and wellbeing. In the present review, the job resources found to buffer the impact of job demands on compassion fatigue included co-worker support, supervisor support and organisational resources and support. In terms of co-worker support, emotional support from colleagues offered predictive validity over and beyond other factors by being strongly and negatively associated with both burnout and secondary traumatic stress. This suggests that shared experiences foster peer-support which could provide an outlet for catharsis. It might also aid in rejuvenating professionals’ drained psychological resources and revitalise vigour for offering therapeutic services to clients. Previous research on the role of work-based social support in preventing burnout in mental health professionals suggests that support from colleagues was related to lower levels of emotional exhaustion and depersonalisation, and higher levels of personal accomplishment [121,122]. A 2012 study conducted on mental healthcare personnel in the UK found that support from colleagues enhanced professionals’ work engagement [123]. Also, meta-analytic evidence exploring the risk and preventive factors for secondary traumatic stress in trauma workers suggests that optimal levels of social support helps in averting the likelihood of experiencing secondary traumatic stress [124].

Support from the supervision or management also plays a key role in impeding the development of compassion fatigue in mental health workers [85,86]. Moral and/or emotional support offered by a supervisor or a manager, usually an experienced professional, might assist mental health professionals in the management of certain job demands associated with compassion fatigue. For instance, a supervisor might share his or her own experiences, which may provide support to novice professionals. Moreover, since increasing age [125,126,127,128] and work experience [129,130] are two of the most frequently reported protective factors against compassion fatigue, listening to an experienced professional might afford effective coping strategies to mental health professionals vulnerable to compassion fatigue. For instance, in a sample of 107 Licensed Master’s Social Workers (LMSWs), Quinn, Ji and Nackerud [131] demonstrated that LMSWs who positively appraised clinical supervision experienced fewer symptoms of secondary traumatic stress. In the two studies included in the present review which investigated the role of supervision [85,86], the relationship of frequency and quality of supervision were not examined simultaneously with compassion fatigue. A study on Italian mental health professionals by Cetrano and colleagues [86] explored the contribution of trust between the professionals and the managers, but not the frequency of contact. In contrast, in the study by Bell, Hopkin and Forrester [85], prison mental health staff were asked to report only the frequency of regular supervision and support from management on a continuous measure. A comparison of effect sizes showed that the frequency of support was more strongly associated with compassion fatigue than the quality of support. However, this warrants further research.

Lastly, the availability and use of organisational resources and support emerged as the most commonly reported job resource inoculating mental health professionals against the detrimental effects of compassion fatigue [76,77,78,80,82]. It endows professionals with essential skills and competencies vital for working effectively with sensitive cases or clients experiencing psychological trauma and ensuring that, at the same time, it does not incapacitate mental health professionals. Two empirical investigations on social workers involved in the recovery and counselling efforts for 9/11 terrorist attacks in New York City, US demonstrated that the provision of information to work with the victims of the attack or bereaved clients was conducive in protecting the personnel against burnout and secondary traumatic stress [76,78]. However, the data for the studies were retrospective, i.e., collected almost three years after the tragic incident, making it vulnerable to recall bias.

In conjunction with the provision of essential information, professionals’ constructive use of that information also plays a critical role in circumventing the advancement of compassion fatigue. The use of evidence-based practices has been found to be negatively associated with compassion fatigue [80,82]. However, some evidence examining it has produced mixed findings. For instance, training and current practice in cognitive-behavioural therapy has been found to be positively associated with burnout in psychotherapists [89]. In contrast, advocacy for working through trauma has been negatively related to secondary traumatic stress [82]. This suggests that vulnerability to the components of compassion fatigue comprises a complex interplay of nature and adoption of empirically informed practices aimed at safeguarding mental health professionals.

The present review also found that compassion fatigue was negatively associated with compassion satisfaction [81,83,84,88,90]. Thus, the job resources associated with lower levels of compassion fatigue were also found to be associated with higher levels of compassion satisfaction [82,83,84,85,86,89]. This finding is consistent with the tenets of the JD-R model [92] which state that burnout and work engagement are two ends of a spectrum, and that job resources which enhance employees’ levels of work engagement are also effective in reducing burnout [92]. However, it is imperative to highlight that the degree of the bivariate relationship between compassion satisfaction and secondary traumatic stress [81,83,84,90] remains low in contrast to the moderate degree of relationship between compassion satisfaction and burnout [81,83,84,88]. This is a pivotal finding for research examining the efficacy of interventions aimed at enhancing compassion satisfaction or reducing compassion fatigue. Whilst burnout and secondary traumatic stress are related to one another, they are two independent constructs that vary in terms of their association with compassion satisfaction. Therefore, whilst an intervention intended to escalate compassion satisfaction might be successful in reducing burnout, its likelihood for reducing secondary traumatic stress remains low.

The lack of theoretically driven research regarding the impact of job demands or job resources on compassion fatigue in mental health professionals also has important implications. Since theory-driven research accords a more comprehensive and empirically-sound understanding of a phenomenon [132], its dearth inhibits the testing of more accurate hypotheses or research questions by failing to explain the interactions among the forces causing the outcome(s) under investigation. For instance, in a cross-sectional study on British psychotherapists, working alliance with clients was found to enhance compassion satisfaction and reduce burnout [89]. The researchers [89] explained the contribution of this alliance as, “the channel through which the therapist experiences positive psychological changes in grappling vicariously with the suffering and distress of his or her clients” (p. 399). Although succinct, this explanation could have been enriched with the aid of organismic valuing theory, which suggests that the accommodation of negative cognitive schemas (of the client) into one’s world-view could foster postadversarial growth (compassion satisfaction, in this case) despite negative wellbeing (or burnout) [133]. This shows that positioning a finding in the context of a theoretical model offers a framework for its surmised description which could further be used for guiding applied research.

Only four research studies [76,77,83,87] included in the present review formally stated a theoretical rationale underlying their proposed research questions or models. Some studies [80,86,88,90] mentioned Stamm’s Professional Quality of Life Framework [91], but only to elucidate the concept of compassion fatigue and not to provide a justification for their research questions or hypotheses. This is a concerning finding, because theory-driven research not only advances our understanding of complex interactions of systems causing a phenomenon [134,135], but also facilitates the development and delivery of appropriate and effective interventions aimed at palliating or aggravating the impact of a phenomenon [136]. Therefore, a scarcity of theoretically-driven research endangers the design and implementation of informed and systematic interventions which can further imperil the quality of services provided by mental health professionals. On the other hand, an opposing view in the literature advocated by some practitioners and applied researchers is that research aimed at filling gaps in a theory is tantamount to “advancing theory for theory’s sake, rather than theory for utility’s sake.” [137] (p. 22) This suggests that nontheory-driven research could also make useful contributions to practice. A comprehensive review of the debate between promoters and opponents of theory-guided research is beyond the scope of this article. For more details, readers are directed towards a commentary by Corley and Gioia [137].

It is vital to highlight here that the studies included in the review varied in their methodological quality, with the majority of studies [76,77,82,89,90] not meeting high standards of empirical rigour. Only four studies were rated as “good” [81,86] or “very good” [80,84]. In addition, all the studies included in the present review adopted a cross-sectional design which inhibits the establishment of causal relationships and temporal order [138], and is vulnerable to common-method variance bias [139]. This is a critical finding, as compassion fatigue has been found to compromise the physical and psychological health and wellbeing of mental health professionals. Therefore, a dearth of high-quality empirical research exploring the role of work-related factors is likely to mask our understanding of the true impact of compassion fatigue on the mental health of mental health workers. Furthermore, it could also hinder the exploration to pathways to alleviate its consequences on the therapeutic services offered by mental health professionals to their clients.

### 4.1. Implications and Directions for Further Research

The findings of the current review suggest that work-related factors have the potential to contribute to the development of compassion fatigue in mental health professionals. Hindering job demands (such as workplace trauma, workload and challenging therapeutic settings) appear to act as catalysts in the advancement of this psychological strain. In contrast, evidence suggests that job resources (such as support from co-workers and supervisors, and organisational resources and support) have the capacity to alleviate or halt its progress. This implies the need to provide essential emotional and/or informative resources to assist the development of mental health professionals’ personal capacities to help them cope with the emotionally and cognitively taxing nature of their profession. The positive relationship between the affective support offered by the supervisor or the manager and compassion satisfaction implies the need for providing soft-skills training to supervisors and/or managers to support them in their nurturing role. In addition, the beneficial role of emotional support from colleagues suggests that organisations should promote collaboration between employees or undertake team-building initiatives to foster supportive relationships among employees. It is likely to foster peer support, but with the caveat that it could also endanger the confidentiality of clients’ information. Confidentiality of a client’s case is one of the legal and ethical pillars of mental healthcare [140], and increased interaction with co-workers could risk violation of this principle. Therefore, an organisation must be cautious while reinforcing peer interaction and support among mental health professionals.

The present review also contributes to the scientific research on compassion fatigue in mental health professionals. According to the 2019 annual report of Health and Safety Executive (HSE), the average prevalence of work-relates stress, depression or anxiety in human health and social work sector was 2120 cases per 100,000 workers [141]. Workload, lack of managerial support and experiences of violence at work were some of causes of stress identified in the report [141]. The same causal factors have also been identified in the present review as being related to compassion fatigue. Moreover, compassion fatigue has also been associated with stress, anxiety, depression and health impairment in previous research [11,28,29,77]. However, no scientific investigation in the literature has synthesised the work-related factors associated with compassion fatigue in mental healthcare professionals. The present review, to the best of our knowledge, is the first systematic review to collate and critically analyse the existing evidence exploring the occupational covariates of compassion fatigue in mental health workers.

The review uncovered several methodological flaws in the literature obviating the progress of research and its application in practice. One of the major gaps discovered was a dearth of longitudinal studies exploring the relationship between work-related factors and compassion fatigue. Since cross-sectional research only indicates covariance between two or more variables, its findings cannot be used for establishing causal relationships or temporal order [138]. For instance, encounters with traumatised clients were found to be associated with burnout and secondary traumatic stress by some studies included in the review [77,81,88]. It suggests a relationship between the two variables, but leaves various important questions unanswered such as: *What is the duration of the mental impact of therapeutic work with traumatised clients? Does its effect wane over time or not? What percentage of traumatised clients on the caseload is perceived as exhausting by mental health professionals?* And *How do workload and the therapeutic orientation of professionals interact with each-other in predicting compassion fatigue?* Future research adopting cross-lagged panel designs with more than two waves of data collection would be valuable in addressing some of these questions [101,102,103]. Another reason for advocating longitudinal research is that, by observing the relationships among variables over a period of time, it aids in the clarification of relationships found by cross-sectional approaches [142].

The lack of use of standardised, multiple-item scales to assess workplace trauma was also one of the major gaps identified in the literature here. Despite workplace trauma being the mostly widely studied job demand [77,78,81,85,86,87,88], no included study used a standardised scale, checklist or questionnaire to quantify it. Instead, it was measured using a single item (yes/no) or a nonstandardised, four-item scale in one study [88]. This could hinder its accurate measurement and observed relationships with the components of compassion fatigue. Although, single-item measures are becoming increasingly popular in research for the assessment of various psychological conditions, they are often viewed with scepticism [143]. Empirical evidence examining their efficacy has yielded mixed findings, with some studies arguing that they demonstrate adequate and, in some cases, even better estimates of reliability and validity, in contrast to multiple-item measures [144,145,146,147]. At the same time, others opine that single-item measures are unsuitable for measuring complex constructs [148] and exhibit lower levels of internal consistency reliability and convergent validity when compared with studies using measures with multiple items [144]. A comparison of the predictive validities of workplace trauma measured using a single- or a multiple-item scale in predicting compassion fatigue is a question for future research.

Further, no study included in the present review used the latest, up-to-date normative benchmarks [104] for classifying the participants into different categories. The latest cut-off scores that were based on a review of 30 studies [104] differ from the original cut-off scores provided by Stamm [91]. Also, ProQOL 5 scale [91] was standardised on psychotherapists and, therefore, the use of its cut-off criteria for other healthcare professionals might neglect the differences among varying professional groups. For instance, the level of threshold for burnout and/or secondary traumatic stress might differ greatly between psychiatric nurses and counsellors owing to the differences in the nature of their caseloads. This suggests that the continued use of the original normative data could obscure the prevalence rates of the components of compassion fatigue in different occupational groups. Future empirical research is required to develop new norms for the ProQOL 5 scale [91] for different professional groups.

### 4.2. Limitations

Despite having a rigorous search strategy, the findings of this systematic review must be interpreted within the context of its potential limitations. The scope of the review was limited by the linguistic restriction imposed on the included studies. Inclusion of studies published in other languages and from a diverse set of national and international contexts could have supplemented the findings of the review in terms of their applicability in varied settings. In addition, the exclusion of unpublished manuscripts and/or masters or doctoral theses narrowed the capacity of the review in terms of capturing the broad extent of empirical evidence and coherently synthesising that evidence. Lastly, a pertinent limitation of the present review relates to a statistically nonsignificant estimate of inter-rater reliability observed for the quality assessment of included studies. The intraclass correlation exhibited a nonsignificant, moderate degree of agreement amongst the researchers, reflecting biases in the evaluation of the methodological quality of studies. This could be attributed to biases in the screening and study selection processes adopted for the present review. According to Belur and colleagues [56], inter-rater reliability should be determined at each and every stage of a systematic review to ensure transparency and replicability. In the present review, although the protocol was peer-reviewed, each stage in the PRISMA cycle [62] was carried out independently by a single reviewer (JS). This could have introduced researcher biases, further leading to discrepancies in the scores assigned to studies for their methodological quality. Future systematic reviews of the evidence on compassion fatigue in mental health professionals should control for the above-mentioned biases.

## 5. Conclusions

The present review is the first systematic review to collate the job demands and resources associated with compassion fatigue in mental health professionals. It adopted an organised, scientific approach to review the literature. The results of this review suggest that job demands found to be associated with compassion fatigue included experiences of workplace trauma, workload and therapeutic settings. The job resources found to prevent compassion fatigue were support from co-workers and supervisors and the provision of organisational resources and support. In addition, the findings indicated that job resources mitigated the impact of job demands on compassion fatigue. The findings of the review have implications for both practice and theory, by exposing critical gaps in the literature and by elaborating the practical implications of the results. It could act as a guide for mental healthcare organisations or institutions to refine job demands or enhance resources to safeguard the mental health and wellbeing of their staff. Although the limitations of this review could have influenced the study selection and data extraction processes, which could have impacted the findings, our conclusions are in line with the wider occupational literature on workplace factors and stress. Nevertheless, further research is required to determine the prevalence of compassion fatigue among mental healthcare workers, and to more comprehensively identify the job demands and job resources associated with it.

## Figures and Tables

**Figure 1 ijerph-17-06987-f001:**
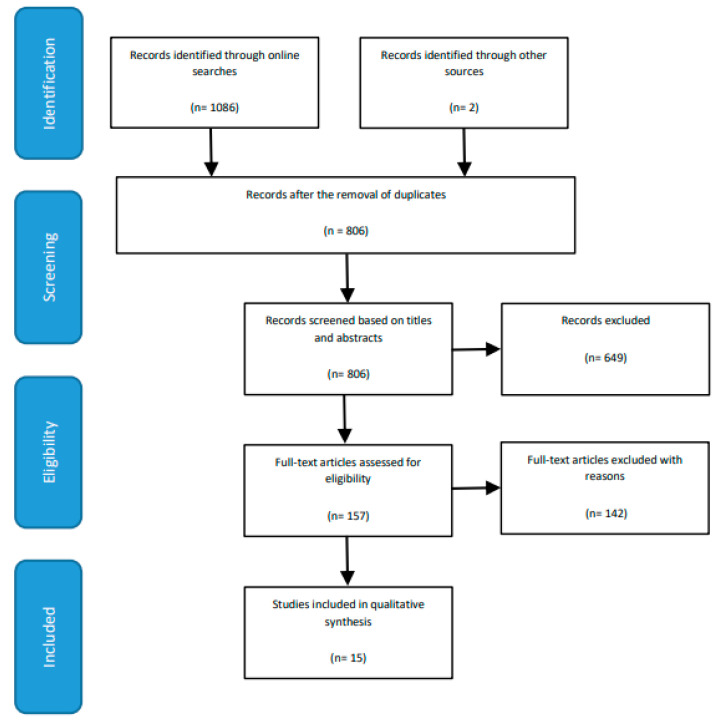
PRISMA Flow Diagram illustrating the steps taken in conducting the systematic review.

**Figure 2 ijerph-17-06987-f002:**
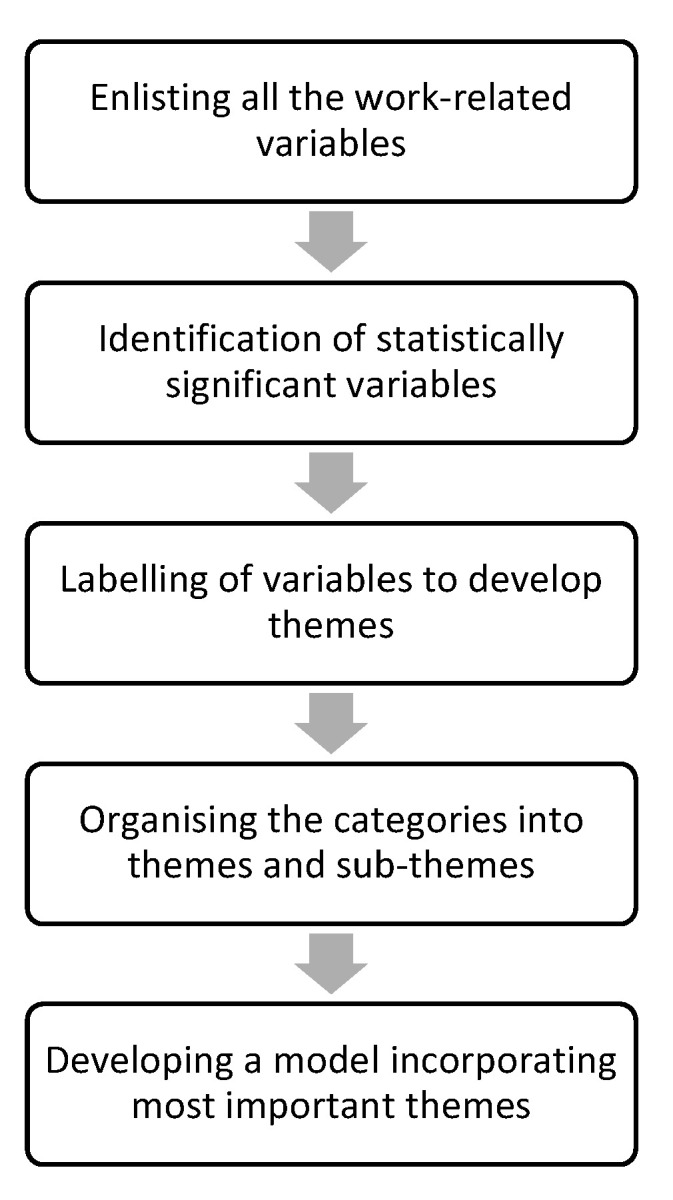
Adapted Coding Process in Inductive Analysis.

**Figure 3 ijerph-17-06987-f003:**
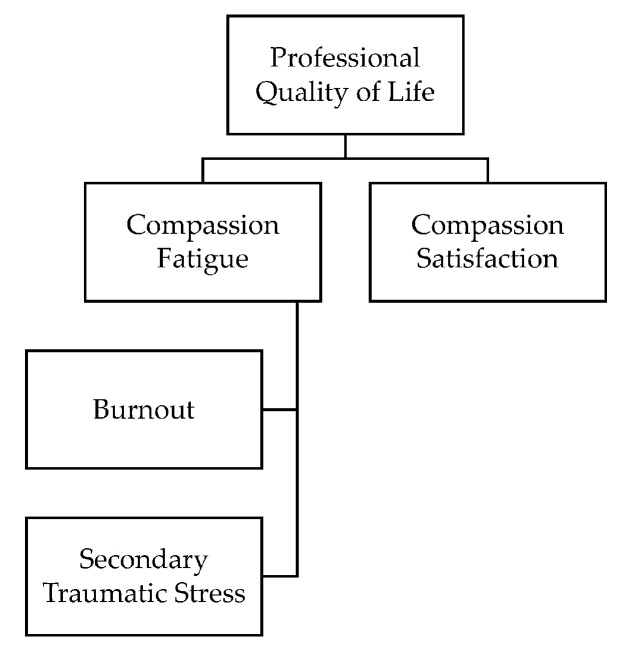
Professional Quality of Life Framework.

**Table 1 ijerph-17-06987-t001:** Study characteristics and CCAT scores.

Study	Country	Study Population	Sample	Aim(s)	Measure	Theoretical Framework	Results	CCAT Score
Adams et al. [76]	USA	New-York based members of the National Association of Social Workers (NASW)	*n* = 236 Males = 20.5% Females = 79.5%	To determine the psychometric properties of the CF (CF) Scale-Revised	CF (CF) Scale-Revised	Stress-Process Framework	Information to work effectively and sense of mastery were negatively associated with BO and STS.	65%
Jacobson [77]	USA	Members of Employee Assistance Professional Association (EAPA)	*n* = 325 Mean Age = 50.06 (8.67) Males = 44.0% Females = 55.7% Involved in clinical work = 47.7% Involved in administrative work = 23.7% Number of clinical hours per week = 21.67 (14.35) Number of EAP sessions = 4.8 (2.96)	To explore the prevalence and predictors of CF, BO and compassion satisfaction in a sample of employee assistance professionals.	ProQOL	Constructivist Self-Development Theory	Work-related stress due to engagement with traumatised clients was positively associated with CF and BO. Services offered by employee assistance professionals to cope with work-related stress was associated with BO.	65%
Boscarino et al. [78]	USA	New-York based members of the National Association of Social Workers (NASW)	*n* = 236 Males = 20.5% Females = 79.5%	To assess the prevalence of CF among social workers who cared for victims of the September 11 attack in New York City.	CF Scale-Revised	NR	World Trade Centre Recovery Involvement was positively associated with STS. Work Environment Support was negatively associated with STS and BO.	70%
Tosone et al. [79]	USA	Manhattan-based members of the National Association of Social Work (NASW)	*n* = 481 Mean Age = 59.83 (9.3) Males = 19.5% Females = 79.6% Length of service = 26.35 (9.77) Psychoanalytic = 61.3% Integrative/Eclectic = 23.7% Cognitive-Behavioural = 6.2% Family Systems = 2.9% General Systems = 2.3%	To explore the relationships among attachment style, resilience and CF.	ProQOL-IV	NR	Percentage of clients experiencing trauma predicted CF.	68%
Craig & Sprang [80]	USA	Clinical psychologists and clinical social workers	*n* = 532 Clinical Psychology = 225 Clinical Social Work = 235 Mean Age = 53.2 Males = 34% Females = 65%	To examine the association between the use of evidence-based practices and CF, BO and compassion satisfaction.	ProQOL-III	ProQOL Framework	Utilisation of evidence-based practices reduced CF and BO.	88%
Butler et al. [81]	USA	Students in the graduate social work training programme at the University of Buffalo	*n* = 195 Males = 11.8% Females = 87.7% Mental health or substance abuse = 31.1% School social work = 20.4% Child welfare = 19.9% Organisations/Community = 12.2% Healthcare = 9.2% Trauma programme/Domestic violence = 8.2% Crisis intervention = 5.1% Residential treatment = 3.1%	To examine trauma-related exposures in graduate training and to investigate whether training-related risk and protective factors predict BO, decline in health status, STS and compassion satisfaction.	ProQOL 5	NR	Training retraumatisation, high field stress and decreased self-care effort were predictors of BO and STS.	83%
Deighton et al. [82]	Germany, Austria and Switzerland	German speaking trauma therapists based	*n* = 100 Males = 34% Females = 65% Clinical Psychology = 35% Other branches of Psychology = 13% Psychiatry = 10% Other clinical professionals = 9% Social workers = 18% Physiotherapy = 6% Art Therapy = 7% Child Therapy = 1% Psychoanalysis/Psychodynamic therapy = 15% Systemic Family Therapy = 12% Cognitive-Behaviour Therapy = 11% Client-Oriented Therapy = 9% Creative Therapies = 9% Gestalt Therapy = 8% Integrative therapy = 5% EMDR = 3% Psychodrama = 2% Other therapies = 8% Number of colleagues = 5.9 (3.5) Weekly working hours = 24.7 (11.2) Weekly caseload = 10.4 (5.4) Length of service with traumatised clients = 7.7 (5.5)	To explore the nature of relationship between trauma therapists’ advocacy and use of working through and work-related outcomes.	ProQOL-III	NR	The degree of working through was negatively associated with BO. The advocacy of working through was positively was positively associated with CF. The participants who advocated working through but did not practice it experienced higher levels of CF than those who advocated as well as practiced it.	53%
Somoray et al. [83]	Australia	Mental health workers working in an NGO providing counselling services	*n* = 156 Mean Age = 44.60 (12.42) Males = 17.9% Females = 79.5%	To investigate the role of personality traits and workplace belongingness in predicting the ProQOL.	ProQOL 5	Five Factor Model of Personality	Workplace belongingness was positively associated with BO.	70%
Mangoulia et al. [84]	Greece	Registered and assistant psychiatric nurses in 12 public hospitals	*n* = 174 Registered nurses = 51.2% Assistant nurses = 48.8% Mean Age = 36.87 (7.37) Males = 29.9% Females = 70.1% Length of service in psychiatric unit = 6.71 (6.65)	To investigate the prevalence of CF, BO and compassion satisfaction and examine the personal and work-related factors associated with them.	ProQOL-IV	NR	Participants who considered their working environment as very good and that the staff always worked as a team reported lower levels of BO. Nurses who worked fewer weekends per month had 1.2 to 2.6 times higher levels of CF.	88%
Bell et al. [85]	UK	Mental health nurses and correctional officers employed at a prison in London	*n* = 36 Mental health nurses = 21 Correctional officers = 15 Mean Age = 40.31 (1.57) Males = 58.3% Females = 41.7%	To determine the levels of CF, BO and compassion satisfaction; and, to explore the relationship between risk and protective factors and ProQOL.	ProQOL 5	NR	Low CF was associated with support and consultation from line managers, emotional support from colleagues and perceived level of skills. Low BO was associated with supervision and emotional support from colleagues.	75%
Cetrano et al. [86]	Italy	Mental health professionals based in three mental health institutions	*n* = 400 Males = 24.1% Females = 75.9% Psychiatry/Training = 19.9% Psychiatric Nursing = 30.5% Psychology = 6.8% Education/Social Work = 16.4% Rehabilitation Therapy = 5.5% Support Work = 20.9%	To examine the predictive association of quality of working life and ProQOL.	ProQOL -III	ProQOL Framework	Ergonomic problems and impact of work on life were associated with CF and BO. Impact of life on work was associated with CF. Trust and perceived risks for future were associated with BO.	80%
Verhaeghe et al. [87]	Belgium	Nursing staff employed at 17 wards in psychiatric hospital	*n* = 219 Mean Age = 41.23 (11.43) Males = 23.7% Females = 72.6%	To explore the associations between attitudes and perceived self-efficacy toward aggression and nurse-related characteristics.	ProQOL 5	Theory of Planned Behaviour (Fishbein and Ajzen, 2010) Bandura’s theory of self-efficacy (Bandura, 1991)	STS was negatively correlated with staff confidence. BO was negatively correlated with prediction and staff anxiety and fear of assault.	78%
Itzhaki et al. [88]	Israel	Nurses working at a mental health centre	*n* = 177 Males = 11.86% Females = 87.01%	To explore the relationships among workplace violence, job stress and ProQOL (ProQOL).	ProQOL 5	ProQOL Framework	Work stress was positively associated with BO.	70%
Linley & Joseph [89]	UK	Psychotherapists	*n* = 156 Mean Age = 53.67 (10.90) Males = 21.79% Females = 78.21% Weekly caseload = 12.64 (6.60)	To examine the impact of organisational level factors and psychological level factors on the positive and negative well-being of psychotherapists.	ProQOL	NR	Therapists who had either received or were receiving personal therapy reported less BO. Therapists with cognitive-behavioural and existential therapeutic orientation were more likely to experience BO.	55%
Ray et al. [90]	Canada	Frontline mental health professionals (FMHPs) in South-western Ontario	*n* = 169 Mean Age = 43.8 (11.61) Males = 18.3% Females = 81.7% Length of service in the profession = 17.23 (11.45) Length of service in mental health = 13.98 (9.86) Length of service in current setting = 6.67 (7.02) Caseload = 14.30 (16.10) Nursing = 40.8% Allied Healthcare = 17.2% Case Management = 17.8% Mental Health = 24.3%	To explore relationships among compassion satisfactions, CF, work-life conditions and BO.	ProQOL-IV-R	ProQOL Framework and Compassion Stress/Fatigue Model	CF was negatively correlated with workload, control, reward, community and fairness.	68%

Note. ProQOL: Professional Quality of Life; BO: Burnout; STS: Secondary Traumatic Stress; CF: Compassion Fatigue.

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
