# Peer review of "A Systematic Review of Job Demands and Resources Associated with Compassion Fatigue in Mental Health Professionals"

_ijerph, 2020, doi:10.3390/ijerph17196987_

Round 1

Reviewer 1 Report

Dear Authors.

This looks like a very interesting review, I found the manuscript informative. Additional Comments for "A Systematic Review of Job Demands and Job Resources Associated with Compassion Fatigue in Mental Health Professionals" are:

Introduction:

This part is very complete and full of information. The way to connect "burn out" and "fatigue compassion" is very clear. But I have a very big doubt about Intro, because most of reviewed studies are cited in this part. It´s not very academic to use the central part of research to justify the whole research.

Methods:

The definition of PRISMA and methodology is very well done.

Lines 184-185 have got a blank space, it´s a formal mistake.

Results:

"3.2. Characteristics of Included Studies", is not clear what part between lines 197-211 are Results or Methods, because there is a description of studies, characteristics of studies, description of participants and characteristics.

Clearly,  "Figure 2. PRISMA Flow Diagram illustrating the steps taken in conducting the systematic review" has to be included in Methods, not in Results.

Table 1 (Study characteristics and CCAT scores) has to be redefined, it´s not acceptable for publication in that way (so many pages for one table!).

"3.6. Qualitative Synthesis of Themes", especially the description of JD-R modelo has to be included in Discussion, not in Results.

3.6.2 Figure 5 has got same information than lines 346-349. Information has to be in text or in figure, not in both. 

In Results part, when you make a systemativ review it´s normal to include references to reviewed studies. But there are several references and texts more connected with Discussion than with Results. Please, review the part.

4.2 Limitations: I love Oscar Wilde. I love Ernst. I think it´s not very academic the reference to Oscar Wilde, even in Limitations

Conclusions: lines 734-739 have got common information, not exactly about the review. 

Author Response

Hello,

Thank you for your valuable feedback. The authors have made the changes suggested by you regarding the studies in the introduction, the inclusion of PRISMA flow diagram in the methods section, the dimensions of the table, and the details provided in the conclusion. The authors have also provided a clarification regarding inclusion of a handful of external references in the results section.

Regards,

Jasmeet Singh

Maria Karanika-Murray

Thom Baguley

John Hudson

Reviewer 2 Report

The review is very interesting and novel. The methodology used by the authors is presented in a detailed and rigorous way. This scientific work is useful because it summarizes all the usual aspects that can contribute to generate relevant stress factors for mental health professionals (workload, workplace trauma, and therapeutic settings). The theoretical background is full of theoretical references that adequately illustrate the constructs investigated. The results can be used to plan burnout prevention strategies for these workers. I only have a few suggestions for the authors.

  • Please, format table 1 to make it easier to read. Specifically, adjust the columns related to study population, aims, and results.

  • I suggest enriching the paragraph of conclusions by making explicit and clear the implications that the results of the summarized studies could bring in the work contexts. For example: what conclusions do the authors reach? It should be repeated briefly in this paragraph.

Author Response

Hello,

Thank you for your valuable feedback.

The modifications regarding the dimensions of the table and the details provided in the conclusion section have now been made.

Regards,

Jasmeet Singh

Maria Karanika-Murray

Thom Baguley

John Hudson

Reviewer 3 Report

The paper is a PRISMA based review on compassion fatigue, i.e. secondary traumatic stress and burnout on mental health workers. The review is well organized and methodologically appropriate, and the existing literature is well organized and synthetized. Works factor influencing compassion fatigue are analytically examined.

The search strategy and inclusion and exclution criteria are clearly explained, and also if the literature search is limited only to English papers, this is stated as a limit in the limits sections. Eligible papers were evaluated with the Crowe Critical Appraisal Tool and rank score given. Interrater agreement for this score was calculated as well as intra-class correlation.

The strategy for data synthesis is explained.

The results show that burnout is the most frequent dimension of compassion fatigue on mental health workers. It seems that there is a shared variance between secondary traumatic stress and burnout, with practical consequences in organizing and managing work in this class of professionals. Limits of the studies are explained and researchers may use the paper to ameliorate future research protocols. Based on a mixed inductive-deductive research strategy the Authors propose some considerations on the development of secondary hand-trauma which may have practical consequences in supervising mental health workers in clinical settings with traumatic patients.

Another relevant point of this review concerns the importance that the availability of resources for the prevention of emotional fatigue, indicating a negative correlation between availability of resources and emotional fatigue. Suggestions are also given on how to develop better research by orienting it theoretically in a coherent form.

I have found the paper relevant for synthetizing a scarcely studied issue. The major limitations are due to the low numbers of papers in the field, so that many considerations that the authors bring may be not fully justified. Still, in my opinion, at least organize the literature in a synthetic way.

Author Response

Hello,

Thank you very much for your valuable feedback.

Regards,

Jasmeet Singh

Maria Karanika-Murray

Thom Baguley

John Hudson